# First Identification and Pathogenicity Evaluation of an EV-G17 Strain Carrying a Torovirus Papain-like Cysteine Protease (PLCP) Gene in China

**DOI:** 10.3390/v15081747

**Published:** 2023-08-15

**Authors:** Zhan-Hong Li, Zhuo-Ran Li, Pei Zhu, Zhen-Xing Zhang, Jian-Ling Song

**Affiliations:** 1Yunnan Tropical and Subtropical Animal Virus Diseases Laboratory, Yunnan Animal Science and Veterinary Institute, Fengyu Road, Jindian, Panlong District, Kunming 650224, China; dy081lzh@163.com (Z.-H.L.); lizhuoran85@126.com (Z.-R.L.); zpcau@sina.com (P.Z.); zhenxing978@163.com (Z.-X.Z.); 2Key Laboratory of Transboundary Animal Diseases Prevention and Control (Co-Construction by Ministry and Province), Yunnan Animal Science and Veterinary Institute, Fengyu Road, Jindian, Panlong District, Kunming 650224, China

**Keywords:** enterovirus G, genetic recombination, papain-like cysteine protease, pathogenicity, host innate immune response

## Abstract

Enterovirus G (EV-G) is prevalent in pig populations worldwide, and a total of 20 genotypes (G1 to G20) have been confirmed. Recently, recombinant EV-Gs carrying the papain-like cysteine protease (PLCP) gene of porcine torovirus have been isolated or detected, while their pathogenicity is poorly understood. In this study, an EV-G17-PLCP strain, ‘EV-G/YN23/2022’, was isolated from the feces of pigs with diarrhea, and the virus replicated robustly in numerous cell lines. The isolate showed the highest complete genome nucleotide (87.5%) and polyprotein amino acid (96.6%) identity in relation to the G17 strain ‘IShi-Ya4’ (LC549655), and a possible recombination event was detected at the 708 and 3383 positions in the EV-G/YN23/2022 genome. EV-G/YN23/2022 was nonlethal to piglets, but mild diarrhea, transient fever, typical skin lesions, and weight gain deceleration were observed. The virus replicated efficiently in multiple organs, and the pathological lesions were mainly located in the small intestine. All the challenged piglets showed seroconversion for EV-G/YN23/2022 at 6 to 9 days post-inoculation (dpi), and the neutralization antibody peaked at 15 dpi. The mRNA expression levels of IL-6, IL-18, IFN-α, IFN-β, and ISG-15 in the peripheral blood mononuclear cells (PBMCs) were significantly up-regulated during viral infection. This is the first documentation of the isolation and pathogenicity evaluation of the EV-G17-PLCP strain in China. The results may advance our understanding of the evolution characteristics and pathogenesis of EV-G-PLCP.

## 1. Introduction

Porcine enteroviruses (PEVs), which are members of the *Picornaviridae* family, were originally divided into 13 serotypes (PEV-1 to PEV-13) on the basis of virus neutralization assay results and were then subdivided into three genera, including *Teschovirus* (PEV-1 to PEV-7 and PEV-11 to PEV-13), *Sapelovirus* (PEV-8), and *Enterovirus* (PEV-9 and -10) based on genomic analyses [1,2]. In order to be distinguished from the enteroviruses (EVs) that infect humans (species A–D), cows (E and F), and non-human primates (A, B, D, H, and J), the EVs that infect pigs have been reclassified as species G (EV-G) [3]. The genome of EV-G is constituted by positive-sense, single-stranded RNA that is 7400–7500 nucleotides (nt) in length and consists of only one large open reading frame (ORF) flanked by a 5′ untranslated region (UTR) and 3′-UTR with a poly(A) tail [1,4,5]. The single ORF encodes a polyprotein that can be proteolytically cleaved to produce four structural capsid proteins (VP1, VP2, VP3, and VP4) and seven nonstructural proteins (2A^pro^, 2B, 2C^ATPase^, 3A, 3B, 3C^pro^, and 3D^pol^) [4,5]. The VP1 protein is located at the surface of the virion and is exposed to the host’s immune system; therefore, the VP1 gene shows the highest sequence variability when compared to the other parts of the genome. Currently, 20 genotypes (EV-G1 to EV-G20) have been identified, according to the divergence of the VP1 gene nucleotide sequence (>25% divergence between different genotypes) [3,6,7].

Genomic RNA recombination is a major driving force of RNA virus evolution [8,9], and the major hallmarks of enteroviral evolution are their high mutation and recombination rates [10,11]. As for the EVs, extensive genetic recombination events occur between the members of the same or different EV species/genera (intratypic recombination), most frequently in the nonstructural genome region (the 2A–2C portion of the enterovirus genome) [12,13,14]. The genetic recombination between EVs with viruses from different families or orders (intertypic recombination) seems to occur much less frequently [15]. Recently, the recombinations between the papain-like cysteine protease (PLCP) gene of toroviruses and the EV-G genotypes G1, G2, G8, G10, G12, and G17 have been observed, and at least two recombinant types (type 1 and type 2) have been identified [16,17,18]. Type 1 recombinant EV-Gs carry the PLCP gene at the junction of the 2C–3A genes of EV-G and were detected in pig feces collected from the USA, Belgium, Germany, and South Korea [17]. In contrast, in type 2 recombinant EV-Gs, the PLCP gene replaces the entire viral capsid protein region up to the VP1/2A junction of EV-G, and has been detected in pig feces collected from China and Japan [16,17]. 

Currently, most studies on EV-Gs have focused on epidemiological surveys [2,5,7,19,20], and there are only a few reports on the pathogenicity of EV-Gs [21,22]. Generally, EV-G infection is asymptomatic in pigs [5], but sometimes it can cause mild disease; for example, the prototypes of EV-G1 (AF363453) and EV-G2 (AF363455) caused skin lesions in naturally infected pigs [23], and the EV-G1 strain CH/17GXQZ/2017 caused flaccid paralysis in an experimental infection of piglets [22]; however, the mechanisms of their pathogenesis remain unclear. The inserted exogenous PLCP gene encodes a deubiquitinase/deISGylase that could potentially suppress the host cellular innate immune responses and might represent a novel host immune control strategy for EV-Gs, but the pathogenic characteristics of the recombinant EV-G-PLCP strains were not verified through the experimental infection of pigs [24].

In the present study, five EV-G nucleic acid-positive samples were detected from 40 pig fecal specimens using EV-G-specific nested RT-PCR [25]; subsequently, one EV-G-PLCP strain was isolated, and then its genetic characteristics and pathogenicity, and the innate immune responses of infected piglets were investigated.

## 2. Materials and Methods

### 2.1. Sample Collection and Virus Isolation

In December 2020, 40 fecal specimens were collected from pigs with diarrhea from a farm in the Yunnan Province, China. Each sample was diluted in phosphate-buffered saline (PBS) to generate a 10% (*w*/*v*) fecal homogenate suspension, followed by the clarification of the suspensions via centrifugation at 3000× *g* for 15 min at 4 °C. The total viral RNA/DNA was extracted from 200 µL suspension using the MiniBEST Viral RNA/DNA Extraction Kit Ver.5.0 (Takara Bio, Dalian, China), according to the manufacturer’s instructions. The nucleic acids of porcine epidemic diarrhea virus (PEDV), transmissible gastroenteritis virus (TGEV), porcine rotavirus A (PRV-A), porcine deltacoronavirus (PDCoV), and EV-G were detected by RT-PCR [25,26] by applying the PrimeScriptTM One Step RT-PCR Kit (Takara Bio). The supernatant fluids of EV-G-positive samples were filtered through a 0.22 µm pore size filter (Millipore, Billerica, MA, USA) and stored at −80 °C until EV-G isolation.

Baby hamster kidney (BHK-21) cells (The China Center for Type Culture Collection, Wuhan, China) were cultured in 25 cm^2^ flasks with Modified Eagle’s Medium (MEM) (Gibco, Invitrogen, CA, USA) containing 10% heat-inactivated fetal bovine serum (FBS) (Gibco), 0.1 mg/mL streptomycin, and 100 U/mL penicillin. At 90% confluency, the cells were washed three times with PBS and incubated with 1 mL filtrates for 1 h; then, the suspensions were discarded, and the cells were washed with PBS once, and 10 mL of maintenance MEM with 1% FBS, 0.1 mg/mL streptomycin, and 100 U/mL penicillin was added and the cells were cultured at 37 °C in 5% CO_2_. The cytopathic effect (CPE) was inspected daily, and three to five blind passages were performed until an obvious CPE was observed. The infected cells were harvested and subjected to three freeze–thaw cycles and were centrifuged at 8000× *g* for 10 min at 4 °C to remove the cell pellets; the supernatants were stored at −80 °C. The viral nucleic acids were extracted from 200 μL virus supernatant, and EV-G-specific RT-PCR [25] was performed to identify the EV-G RNA. 

### 2.2. Virus Purification and Replication Kinetics Analysis

The isolated virus EV-G/YN23/2022 was purified by plaque purification. Briefly, BHK-21 cells were grown in 12-well plates to a 90% monolayer, and then the growth medium was discarded, and the cells were washed three times with PBS; then, 10-fold serially diluted (10^−1^ to 10^−6^) stock virus was inoculated onto the cells (0.4 mL/well) for 1 h at 37 °C. Then, the virus inoculum was discarded, and the plates were washed twice with PBS and covered with 2 mL MEM containing 1% (*w*/*v*) agarose (Sigma-Aldrich, St. Louis, MO, USA) and 1% FBS (Gibco) and incubated at 37 °C. After 96 h of incubation, 1.5 mL 0.01% neutral red staining solution (Solarbio, Beijing, China) was added per well at 37 °C for 5 h. Then, a clear and uniform plaque was picked and used to inoculate new BHK-21 cells in 96-well plates. When approximately 80% of the cells displayed visible CPEs, the cells were frozen at −80 °C and thawed twice. After centrifugation at 8000× *g* for 10 min, the supernatant was subjected to three additional rounds of plaque purification, and the plaque-forming units (PFU) of the fourth purification virus stock was counted.

For the growth kinetics assay, BHK-21, porcine kidney (PK-15) cells (The China Center for Type Culture Collection), swine testicle (ST) cells (The China Center for Type Culture Collection), monkey embryonic kidney epithelial (Marc-145) cells (The China Center for Type Culture Collection), and pig small intestinal mucosal epithelial (SIEC) cells (gift from Prof. Yaojun, Yunnan Animal Science and Veterinary Institute) were seeded into monolayers in 6-well plates and were infected with the purified viruses (multiplicity of infection (MOI) = 0.01). After incubation for 1 h at 37 °C, the cells were washed twice with PBS, and then 3 mL maintenance MEM medium was added. Aliquots (150 μL) of the infected cell culture medium were removed at intervals of 12 h from 12 h post-infection (hpi) to 120 hpi, and the equivalent fresh maintenance medium was added immediately after sampling. The virus titers in the cell culture medium were determined using BHK-21 cells and the previously described standard plaque assays, with a slight modification: the infected cells were stained with 0.5% crystal violet (Solarbio) in 30% methanol (Xilong Scientific Co., Ltd., Guangzhou, China) for 3 h at room temperature.

### 2.3. Virus Genome Amplification and Sequencing

The nearly complete genome of EV-G/YN23/2022 was obtained from overlap extension RT-PCR with four pairs of primers (Table 1) using the PrimeScript^TM^ One Step RT-PCR Kit (Takara Bio). Briefly, the 50 µL One Step RT-PCR reaction mix contained 25 µL One Step Buffer (2×), 2 µL of PrimeScript One Step Enzyme Mix, 2 µL each of the corresponding primers (20 μM), 5 µL denatured total RNA, and 14 µL RNase-free water. The RT-PCR was performed according to the standard protocol of the kit: 50 °C for 30 min to synthesize the complementary DNA (cDNA), followed by a denaturation step at 94 °C for 2 min, and then 30 cycles of 94 °C for 30 s, 58 °C for 30 s, and 72 °C for 2 min 30 s, with a 7 min extension period at 72 °C. The PCR products were checked by running on a 1.5% agarose gel stained with GoldView II (Solarbio) and visualized using the Molecular Imager^®^ Gel Doc™ XR System (Bio-Rad Laboratories Inc., Hercules, CA., USA). The expected PCR products were purified using a gel extraction kit (Takara Bio) and cloned into the pMD19 vector (Takara Bio). The cloned products were transformed into DH5α competent cells (Takara Bio), and the positive clones were screened using PCR and were then sent to Kunming Shuoqing Biological Technology Company (Shuoqing Bio, Kunming, China) for sequencing. The obtained sequences were assembled using Lasergene software (DNASTAR Inc., Madison, WI, USA). 

### 2.4. Sequence and Phylogenetic Analysis

The reference complete nucleotide and predicted amino acid (aa) sequences of different EV-G genotypes (G1 to G20) were downloaded from GenBank (Appendix A), and the ORF of the isolated EV-G/YN23/2022 strain was found and translated into its amino acid sequence using EMBOSS ORF finder “(http://emboss.bioinformatics.nl/cgi-bin/emboss/getorf accessed on 27 November 2022)”.The nucleotide and predicted amino acid sequences were aligned using MAFFT software [27], and the identity values of the sequences were calculated using BioEdit (Version 7.1.3.0) [28]. Phylogenetic trees were generated by the maximum likelihood (ML) method using MEGA 6.06 software [29]. The appropriate best-fit model (with the lowest Bayesian Information Criterion scores) for the complete genome and VP1 gene sequence was both the general time-reversible model with a gamma distribution with invariant sites (GTR+G+I), and 1000 replicates of the bootstrap value were used to evaluate the reliability of the phylogenetic trees. The potential recombination events in the EV-G/YN23/2022 genome sequence were evaluated using the Recombination Detection Program 4 (RDP4, Version 4.101) [30] and SimPlot software (Version 3.5.1) [31]. The nucleotide sequence reported in the present research is available from GenBank under accession number OQ988093.

### 2.5. Experimental Infection of Piglets

Eight 28-day-old piglets were purchased from a pig farm in Fuyuan County, Yunnan Province, and were randomly divided into two groups (Group A and B) of four pigs, with each group housed in separate barns. All pigs were confirmed to be negative for EV-G, PEDV, TGEV, PRV-A, and PDCoV by virus-specific RT-PCRs [25,26]. After three days of adaptation, the challenged group (Group A) of piglets were orally inoculated with 5 mL of the EV-G/YN23/2022 strain (2 × 10^6^ PFU/mL), and the piglets in Group B received 5 mL of BHK-21 cell culture media supernatant orally as negative controls. The body weights, rectal temperatures, feces status, and other clinical symptoms, for example, skin lesions and flaccid paralysis of the piglets in both groups, were monitored and recorded daily. Whole blood samples were collected at 0, 2, 4, 6, 9, 12, 15, 18, and 21 days post-inoculation (dpi) to detect the cytokine mRNA levels, and the serum samples were collected at the same time points to detect the cytokine concentrations and the antibody titers against EV-G/YN23/2022.

### 2.6. Detection of Cytokine mRNA Levels and Concentrations 

Total RNA was extracted from 500 μL aliquots of each whole blood sample using the RNAprep Pure Hi-Blood Kit (TIANGEN Bio, Beijing, China), following the manufacturer’s instructions, and was quantified using NanoVue Plus (GE Healthcare, Chicago, IL, USA). Then, 1 μg of RNA was reverse transcribed to cDNA using the PrimeScript™ RT reagent kit with gDNA eraser (Takara Bio), following the manufacturer’s instructions. Fifty nanograms of the synthesized cDNA was used as a template to evaluate the gene expression levels of interleukin-1β (IL-1β) [32], IL-6 [32], IL-18 [33], tumor necrosis factor-α (TNF-α) [32], interferon-α (IFN-α) [34], IFN-β [35], IFN-λ3 [35], interferon-stimulated gene 15 (ISG-15) [36], and IFN regulatory factor 7 (IRF-7) [37] via qPCR using the primer sets described previously (Appendix A), and β-actin [34] was used as an endogenous control to normalize the amount of input cDNA. The relative quantification (RQ) of mRNA expression of the target genes was calculated using the 2^−ΔΔCt^ method according to a previous report [38], and the results are presented as fold changes relative to the control group.

The qPCR reaction was performed using TB Green^®^ Fast qPCR Mix Kit (Takara Bio) and a 7500 Fast Real-Time PCR System (Applied Biosystems™, Thermo Fisher Scientific, Waltham, MA, USA), according to the manufacturer’s recommendations. Briefly, the qPCR was performed in a final volume of 20 μL, containing 10 μL TB Green Fast qPCR Mix, 0.8 μL forward/reverse primer (10 μM), 0.4 μL ROX Reference Dye II, 2 μL cDNA template, and 6 μL ddH_2_O. The reaction conditions contained a cycle of initial denaturation at 95 °C for 30 s, followed by 40 cycles of denaturation at 95 °C for 3 s and annealing and extension at 60 °C for 15 s, followed by the melt curve stage: 95 °C for 15 s, 60 °C for 1 min, and 95 °C for 15 s. 

Meanwhile, the concentrations of IL-1β, IL-6, IL-18, TNF-α, IFN-α, IFN-β, and IFN-λ3 in the serum samples were measured using commercial ELISA kits (Meimian, Yancheng, China) and following the manufacturer’s instructions.

### 2.7. Virus Shedding and Tissue Virus Load Detection

In order to detect the viral load, an EV-G-specific SYBR-Green RT-quantitative PCR (RT-qPCR) was developed for this study by using a pair of primers (Table 1) targeting the 5′-UTR of the EV-G/YN23/2022 genome. Furthermore, the amplification product was cloned into the pMD19 vector (Takara Bio) to serve as the standard template and was serially diluted (10-fold) to generate a standard curve for each plate. The primers were evaluated in silico to ensure that there were no cross-reactions with closely related PEVs. 

Fecal samples were collected from 0 dpi to 21 dpi, and all piglets from Group A and Group B were humanely euthanized at 21 dpi; then, the heart, liver, spleen, lungs, kidneys, brain, small intestine, mesenteric lymph nodes, and stomachs were collected for detecting the viral loads by the developed RT-qPCR. Firstly, 0.5 g of tissue or feces sample was homogenized with stainless steel beads of 2 mm in diameter (EASYBIO, Beijing, China) in 1 mL sterile PBS using a TissueLyser II homogenizer (Qiagen, Hilden, Germany) and then centrifuged at 8000× *g* for 10 min at 4 °C. Virus RNA was extracted from 200 μL supernatant using a MiniBEST Viral RNA/DNA Extraction Kit Ver.5.0 (Takara Bio), according to the manufacturer’s instructions, and the RNA was dissolved in 30 μL ddH_2_O. Subsequently, 10 μL of extracted RNA was used to synthesize cDNA using a PrimeScript™ 1st Strand cDNA Synthesis Kit (Takara Bio), according to the manufacturer’s instructions, and reverse transcription was conducted at 30 °C for 10 min, 42 °C for 60 min, and 95 °C for 5 min. Then, the viral RNA copies in the cDNA of different samples were detected through qPCR using the TB Green^®^ Fast qPCR Mix Kit (Takara Bio) and a 7500 Fast Real-Time PCR System (Applied Biosystems™), according to the manufacturer’s instructions. Ten-fold serially diluted (10^7^ to 10^1^ copies/μL) recombinant plasmids of the 5′-UTR of the EV-G/YN/23/2022 genome were used to obtain the standard curves to calculate the number of viral RNA copies in the samples.

### 2.8. Serum Neutralization Antibodies Detection 

The titer of the neutralization antibodies (nAbs) against EV-G/YN23/2022 in the collected serum was detected by a serum neutralization test (SNT), as described previously [39]. Briefly, the serum samples were heat inactivated at 56 °C for 30 min and then two-fold serially diluted with MEM (1:4 to 1:512). One hundred microliters of each diluted sera was added to an equal volume of EV-G/YN23/2022 (100 TCID_50_) in each well of a 96-well plate. After incubation for 1 h at 37 °C, 50 μL aliquots of BHK-21 cells (approximately 2.5 × 10^4^ cells per well) were seeded into all of the wells and were then incubated at 37 °C with 5% CO_2_ for 72 h to 120 h. The CPEs were observed with an inverted microscope, and the neutralization antibody titer of the serum was defined as the highest serum dilution, that inhibited 50% of virus growth compared to the negative serum control.

### 2.9. Histopathological Examination

At 21 dpi, all pigs were necropsied, and the brain, small intestine and mesenteric lymph node were collected and fixed in 4% neutral-buffered formaldehyde (Servicebio, Wuhan, China) for 48 h and then dehydrated, cleared in xylene (Servicebio), embedded in paraffin wax (Sakura, Japan), sectioned (5 μm thick) using a microtome (Shanghai Lycra, Shanghai, China), and mounted onto slides, which were then stained with hematoxylin and eosin (H&E staining) (Servicebio) for histopathological examinations using standard light microscopy (Nikon, Tokyo, Japan).

To detect the distribution of EV-G in the brain, small intestine, and mesenteric lymph nodes, an RNA-based in situ hybridization (ISH) assay was performed according to a previous report [40]. Briefly, the RNA probe, ‘EVG-P’, targeting the 2C gene of EV-G/YN23/2022 (position: 4494–4794), was labeled with DigU (Roche, Mannheim, Germany). The tissue samples were fixed with 4% neutral-buffered formaldehyde (Servicebio) and were embedded in paraffin (Sakura), and sliced to 5 μm thick sections using a microtome (Shanghai Lycra); the sections were deparaffinized in xylene (Servicebio) and a gradient of ethanol concentrations (Servicebio), followed by treatment with 3% hydrogen peroxide (Servicebio) in PBS for 20 min and digestion by 20 μg/mL proteinase K (Servicebio) for 15 min at 37 °C. Pre-hybridization was performed in Hybridization Buffer (50% formamide (*v*/*v*), 900 mM NaCl, 20 mM Tris-HCl, pH 7.2) (Servicebio) with DNA blocking reagent (0.1 mg/mL) (Roche) at 37 °C for 1 h and the sections were subsequently hybridized in Hybridization Buffer with the EVG-P probe (500 nM) overnight at 40 °C. After a serial wash with saline sodium citrate (SSC) buffer (Servicebio) at different concentrations (2 × SSC to 0.5 × SSC), the sections were blocked with 10% (*v*/*v*) rabbit serum (Servicebio) at room temperature for 30 min; the blocking solution was then discarded, and the sections were incubated with the Anti-Digoxigenin-AP (Fab Fragments) antibody (Jackson Immuno Research Inc., West Grove, PA, USA) at 37 °C for 40 min and then washed with Tris-buffered saline Tween-20 (TBST) (Servicebio) four times. The slides were developed in alkaline phosphatase buffer with BCIP/NBT (Boster Bio, Wuhan, China), and the presence of EV-G RNA was inspected through a light microscope (Nikon).

### 2.10. Statistical Analysis

Each sample was tested in triplicate; all the values were presented as the means ± standard deviations (SD), and the statistical analysis was performed using GraphPad Prism 8 (GraphPad Software Inc., San Diego, CA, USA). The differences among the groups were analyzed using one-way analysis of variance (ANOVA), and a probability (*p*) value < 0.05 was considered statistically significant. The significance level for all analyses was set as *****
*p* < 0.05, ******
*p* < 0.01, and *******
*p* < 0.001.

### 2.11. Ethics Statement

In this study, the piglets were fed in barns that allowed free movement and access to food and water, and the animal experiment was conducted in biosafety level 2+ facilities at the Yunnan Biological Pharmaceutical Factory (Kunming, China) under the guidance of the Scientific Ethics Committee of YNASVI01-2022006.

## 3. Results

### 3.1. EV-G Isolation and Identification 

One sample (1/40, 2.5%) and six samples (6/40, 15.0%) were detected as PEDV positive and EV-G positive by RT-PCR [25,26], respectively, and no TGEV-, PRV-A-, and PDCoV-positive products were detected. After three cycles of blind passage on the BHK-21 cells, one virus strain that was isolated from the EV-G-positive samples caused strong CPEs in the inoculated cells (Figure 1A), which was identified as EV-G by RT-PCR [25] and designated as EV-G/YN23/2022. Subsequently, the virus was purified using a plaque purification assay, and clear and uniform plaques were obtained (Figure 1B) after four rounds of purification; the titer of the purified virus stock was 6.6 × 10^6^ PFU/mL. Finally, the cell tropism and proliferation of EV-G/YN23/2022 on BHK-21, Marc-145, PK-15, ST, and SIEC cells were tested. Generally, strong CPEs were observed in all the inoculated cells at 24 to 72 hpi (Figure 1A), and the virus showed the most rapid multiplication rate and the highest viral titer (7.2 × 10^6^ PFU/mL) in the SIEC cells when compared to the other cell lines (Figure 1C), and similar growth curves for the BHK-21, Marc-145, PK-15, and ST cells were observed (Figure 1C).

### 3.2. Analysis of Genomic Sequences

The nearly complete genome sequence of the EV-G/YN23/2022 strain was derived via the combination of four overlapping cDNA fragments (Appendix A); this sequence was submitted to the GenBank database under the accession number OQ988093. The complete genome length of EV-G/YN23/2022 was 8033 nt, including the 5′-UTR (812 nt), 3′-UTR (66 nt), and a single, long ORF (7155 nt) (comprising 12 genes) that ranged from 66 nt (3B gene) to 1383 nt (3D gene) (Table 2). It was noted that EV-G/YN23/2022 contained an extra 642 nt sequence within the 2C–3A junction region, and this sequence showed homology to the PLCP sequence variants that were recently found in the EV-G1 and G17 strains isolated in Japan and the USA (Table 2). 

Sequence comparisons of the whole genome and the polyprotein showed that EV-G/YN23/2022 shared the highest nt/aa identities with the EV-G17 genotype strains, ranging from 85.5/95.3% (KY761948) to 87.5/96.6% (LC549655), and generally shared low similarities with the other EV-G genotype strains, ranging from 62.5/73.1% (G7, MG958646) to 84.5/88.5% (G1-PLCP, MF782664) (Table 2). The nt/aa sequences of the different regions of the EV-G/YN23/2022 genome were further compared with nine other representative EV-G strains. The results in Table 2 showed that the 1D/VP1 gene of EV-G/YN23/2022 displays the highest homology with the G17-PLCP strain (LC549661) (83.1/98.3%) (Table 2) at the nt/aa level, indicating that EV-G/YN23/2022 belongs to the G17 genotype; in contrast, the PLCP gene showed low identities, ranging from 76.5% to 85.8% (nt) and 76.6% to 86.9% (aa) (Table 2). Notably, five regions of EV-G/YN23/2022 exhibited a relatively higher homology at the nucleotide level (91.5–95.6%) and amino acid level (96.6–100%) with MF782664 (G1-PLCP), including the 5′-UTR, and 2A, 2B, 3A, and 3D genes. In addition, the 3B gene of EV-G/YN23/2022 was highly conserved at the amino acid level and exhibited 100% identity with eight representative EV-G strains (Table 2).

### 3.3. Phylogenetic Analysis

To further investigate the genetic relationships between EV-G/YN23/2022 and the other EV-G strains, phylogenetic trees were constructed based on the VP1 gene and the complete genome nucleotide sequences available in GenBank. In the phylogenetic tree of the VP1 gene, all the EV-G strains were distributed into 20 groups (G1 to G20) (Figure 2A), and EV-G/YN23/2022 was clustered into the G17 subtype with five other EV-G isolates from the USA (*n* = 4) and Japan (*n* = 1) (Figure 2A). The same phylogenetic topology was observed in the complete genome nucleotide tree; all the G17 subtype strains were clustered within a separate lineage (Figure 2B), and all the G17 strains were PLCP-inserted strains.

### 3.4. Recombination Analysis

To identify the possible recombination events, SimPlot (Version 3.5.1) [31] and RDP4 (Version 4.101) [30] software were used to detect recombination. The standard similarity plot analysis in SimPlot (Version 3.5.1) [31] of the complete genome of EV-G/YN23/2022 as a separate query revealed that EV-G/YN23/2022 showed closer relatedness to EVG 02/NC CHI/2014 in the regions from positions 1 to 708 and 3383 to 8033 than those of other strains (Figure 3A). However, the sequence between nucleotides 709 and 3382 displayed a disproportionately low level of similarity between EV-G/YN23/2022 and EVG 02/NC CHI/2014 when compared to the other regions (Figure 3A). Additionally, identical evidence was provided by the Boot Scanning analysis using the RDP4 software (Version 4.101) (Figure 3B), and the recombinant event showed a high degree of certainty according to the results of seven detection methods (*p*-values ≤ 4.353 × 10^−11^, Appendix A). The recombination event was further substantiated by constructing a phylogenetic tree using MEGA 6.06 software [29]. An analysis of the concatenated regions before and after the breakpoints (at positions 709 and 3382) indicated that EV-G/YN23/2022 had the closest relationship with EVG 02/NC CHI/2014 (Figure 3C), while in the putative recombinant regions (positions 709–3382), they fell into two different sub-lineages (Figure 3D). In general, the results of the similarity plot, bootstrap scanning analysis, and phylogenetic tree construction suggested that the estimated recombination breakpoints are located at the 708 and 3383 positions of the EV-G/YN23/2022 genome, and EVG 02/NC CHI/2014 was the major putative parent, but no minor putative parent was found.

### 3.5. Clinical Symptoms Observation

To assess the pathogenicity of the recombinant EV-G strain containing the PLCP fragment in pigs, 31-day-old piglets were orally inoculated with EV-G/YN23/2022. The piglets in the control group did not show any clinical symptoms during the entire study period. One of the four inoculated piglets showed a febrile response (39.9 °C) at 4 dpi, and all the challenged pigs had a temporary fever (>39.5 °C) from 6 to 9 dpi (Figure 4A); the febrile peak was at 7 dpi, with average body temperatures of 40.5 °C (Figure 4A). All of the challenged animals showed anorexia from 5 to 8 dpi, and the mean daily weight gain was significantly lower than that of the control group (*p* < 0.05, Figure 4B) at 8 dpi. Meanwhile, four piglets in the challenged group showed mild diarrhea between 6 and 9 dpi (Figure 4C), and the diarrhea symptoms gradually recovered after 12 dpi. It is notable that three of the four piglets in the challenged group showed skin rashes (Figure 4D) from 7 to 11 dpi, and the rashes gradually disappeared at 13 to 15 dpi. Overall, no deaths of challenged piglets occurred, and these data suggest that EV-G/YN23/2022 is mildly pathogenic to piglets.

### 3.6. Gross Pathology and Histopathological Analysis

All pigs were euthanized and necropsied at 21 dpi. No typical gross lesions were observed, except for mild hemorrhaging in the small intestine and mesenteric lymph nodes of the challenged animals (Figure 5). The histopathological analysis of the small intestine showed that the epithelial mucosae experienced local exfoliation and defects (blue arrow), and the gland of the lamina propria was arranged less regularly (yellow arrow) in the infected pigs when compared to the mock-infected group (Figure 6A). In contrast, there were no pathological lesions observed in the mesenteric lymph nodes and the brains of the challenged piglets (Figure 6A). Subsequently, the small intestines, mesenteric lymph nodes, and brain (cerebellum) slices were subjected to in situ hybridization to detect the distribution of viral nucleic acids. As a result, EV-G/YN23/2022 RNA-positive signals (green arrow) were identified in the three examined tissues of the challenged piglets (Figure 6B), and the strongest signal was observed in the cerebellum, while no positive signal could be seen in any of the tissues of the mock-infected group animals (Figure 6B). When taken together, these results indicate that EV-G/YN23/2022 may cause intestinal lesions and penetrate the blood–brain barrier to invade brain tissue in piglets. 

### 3.7. Fecal Virus Shedding and Virus Load in Tissues 

The viral titers of EV-G/YN23/2022 in the fecal swab samples collected from the piglets at regular intervals were detected by the RT-qPCR method developed in this study. The results showed that viral nucleic acid was detectable at 1 dpi with a titer of 10^2.2^ copies/0.1 g (Figure 7A) and then increased rapidly from 2 dpi, peaking at 5 dpi with a titer of 10^6.5^ copies/0.1 g (Figure 7A); subsequently, virus shedding gradually decreased but still remained at a detectable level (10^3.1^ copies/0.1 g) at the end of the experiment (Figure 7A). The viral load of EV-G/YN23/2022 in the various tissues was also tested by RT-qPCR, and the results showed that the virus was detected in the brain (10^3.6^ copies/0.1 g), small intestine (10^3.1^ copies/0.1 g), mesenteric lymph node (10^3.1^ copies/0.1 g), spleen (10^3.0^ copies/0.1 g), and liver (10^2.1^ copies/0.1 g), and no viral RNA was detected in the heart, lungs, kidneys, or stomachs of the infected group (Figure 7B). Viral RNA could not be detected in the fecal swab and organ tissues of the mock-infected group throughout the experimental period.

### 3.8. Antibody Responses of Challenged Piglets 

The nAbs against EV-G/YN23/2022 in the serum of the infected piglets were detected by the SNT. As shown in Figure 7C, the nAbs were detectable at 6 dpi, with a mean titer of 1:16, and then gradually increased from 6 dpi, reaching a peak at 15 dpi, with a mean titer of 1:452. The titers then remained constant at approximately 1:443 until the end of the study. No EV-G/YN23/2022-specific antibodies were detected in the sera collected from the mock-infected group animals throughout the study.

### 3.9. Cytokine Gene mRNA Levels in PBMCs and Concentrations in the Serum of EV-G/YN23/2022-Infected Piglets 

The transcriptional levels of multiple cytokine genes, including IL-1β, IL-6, IL-18, TNF-α, IFN-α, IFN-β, IFN-λ3, ISG-15, and IRF-7, were detected through RT-qPCR, and the RQ value of the control group was set to 1. As shown in Figure 8, the mRNA expression levels of IL-6, IL-18, IFN-α, IFN-β, and ISG-15 were up-regulated during EV-G/YN23/2022 infection, while IL-1β, TNF-α, IFN-λ3, and IRF-7 maintained in relatively constant levels, and no cytokines were significantly reduced. Specifically, IL-6 was significantly up-regulated from 4 dpi to 15 dpi (*p* < 0.05) and peaked (38.46-fold) at 6 dpi; IL-18 increased significantly at 9 dpi (2.83-fold) and 12 dpi (6.12-fold) when compared to the mock-infected group. For IFN-α and IFN-β, the expression levels were elevated from 4 dpi onwards, and the highest transcriptional levels (IFN-α: 9.322-fold; IFN-β: 7.74-fold) were at 9 dpi and 6 dpi, respectively. The transcriptional levels of IFN-β decreased to control levels at 12 dpi, while IFN-α maintained higher levels (1.65-fold, although not significantly) at 21 dpi when compared to the mock-infected group. Additionally, the ISG-15 mRNA levels in the EV-G/YN23/2022-inoculated group were significantly increased at 6 dpi (2.98-fold) and 9 dpi (3.61-fold) and stayed at an invariant level during the other time points tested comparing with those of mock-infected group. 

The concentrations (mean ± SD) of swine cytokines (IL-1β, IL-6, IL-18, TNF-α, IFN-α, IFN-β, and IFN-λ3) in the serum samples of the inoculated and control piglets were determined by ELISA. The results demonstrated that the IL-6 concentration of the EV-G/YN23/2022-challenged piglets increased significantly at 6 dpi and 9 dpi when compared to the control group (*p* < 0.05, Figure 9). Although the differences were not statistically significant, the inoculated pigs showed increased serum levels of IFN-λ3, IFN-α, and IFN-β compared to the control pigs at 12 dpi and 15 dpi (Figure 9). The concentrations of IL-1β, IL-18, and TNF-α did not change significantly, and they maintained a relatively constant level throughout the experimental period compared to the control group (Figure 9).

## 4. Discussion

Currently, 20 genotypes (EV-G1 to EV-G20) of EV-Gs have been reported in Europe, North and South America, and Asia [16,19,41], and most of them (EV-G1 to G4, G6, and G8 to G19) were identified from swine, while G5 and G7 were isolated from sheep, and G20 was found in goats [41]. As it was reported, the G17 genotype had been identified in the USA (*n* = 1), Germany (*n* = 1), and Japan (*n* = 4) [19,24,42], and there were only two genotypes (G1 and G20) reported in China [16,41]. According to the criterion of EV-G genotype classification (complete VP1 nucleotide sequence divergence >25%) [3,7], the EV-G/YN23/2022 isolated in this study can be classified into the G17 genotype because the complete VP1 gene sequence identities of the EV-G/YN23/2022 and G17 strains were 82.5–83.1% (Table 2), which was obviously higher than the identities compared with the other genotype strains (60.5–66.1%) (Table 2). To our knowledge, this is the first report of the isolation of a G17 genotype strain in China. 

At the genome level, EV-G/YN23/2022 showed relatively high sequence identity (85.5–87.5% nt and 95.3–96.6% aa) with the other G17 strains (Table 2), which clustered together within a large clade on the phylogenetic tree, while EV-G/YN23/2022 was located on a slightly independent cluster (Figure 2B). This indicated that EV-G/YN23/2022 had a closer genetic relationship with G17 strains than the other genotypes but was somewhat phylogenetically distant from the other G17 strains. It was noteworthy that the structural protein genes (VP1–VP4) of EV-G/YN23/2022 shared higher identities with the G17 strains than the other genotypes (Table 2), while the other major regions (5′-UTR, 2A–2C, 3A, and 3C–3D genes) were most closely related to the G1-PLCP strain EVG 02/NC CHI/2014 (MF782664) isolated from China (Table 2), which suggests that EV-G/YN23/2022 is a hybrid of the G17 strains and EVG 02/NC CHI/2014 (MF782664).

As expected, recombination breakpoints (at positions 708 and 3383) were detected in the EV-G/YN23/2022 genome with high bootstrap support through recombination analyses (Figure 3A,B). The breakpoint at position 3383 is located in the middle of the 2A region (Figure 3A), which is consistent with the previous reports [2,7,18,42], suggesting that the 2A gene may be a hotspot for recombination events for EV-G. The major putative parent of EV-G/YN23/2022 was EVG 02/NC CHI/2014 (Figure 3A,B), while the minor putative parent was not found (Figure 3A,B). Considering the relatively high identities between EV-G/YN23/2022 and the other G17 strains in the P1 region (VP1–VP4 genes), we posit that EV-G/YN23/2022 is a possible mosaic strain from the recombination between G17 strains and EVG 02/NC CHI/2014 (MF782664). We speculate that the reason why the minor parent is not found is due to the limited numbers of whole genome sequences of the G17 strains in the GenBank database. There are only five (USA = 1; Japan = 4) complete genome sequences of G17 strains available in the GenBank database [24,42], and none of these were identified in China. We believe that as more and more G17 complete genome sequences are deposited on the GenBank database in the future, the minor parent of the EV-G/YN23/2022 strain will eventually be discovered. 

The prototypic EV-G1 (AF363453) and EV-G2 (AF363455) strains were originally isolated from pigs with skin lesions [23]; however, skin symptoms were not reported in the isolation of other EV-G strains [1,18,24,43]. In addition, skin lesions were not observed in two independent EV-G pathogenicity experiments, which were conducted by Yang et al. [21] and Xue et al. [22], respectively. Therefore, it cannot be concluded that EV-G infection will cause skin symptoms in pigs. In this study, a recombinant EV-G strain (EV-G/YN23/2022) carrying a PLCP gene was isolated from the fecal samples of pigs with diarrhea, and the regression infection experiment showed that EV-G/YN23/2022 could induce typical skin lesions in pigs (Figure 4D); this is inconsistent with previous reports [21,22]. Considering that the virus strain (CH/17GXQZ/2017) used in the research of Xue et al. [22] was non-PLCP-inserted EV-G, and the fact that only the 3D gene was sequenced for the strains (CH-PW12 and CH-PW20) in another report [21], we speculate that the PLCP-inserted EV-G share the potential to induce skin lesions in piglets. Admittedly, we cannot rule out that the clinical symptoms were more obvious and induced by the artificial high-dose inoculations when compared to natural infections; therefore, further detailed research should be carried out to confirm this speculation. At the same time, it should be noted that the skin symptoms lasted for a relatively short time in our study, and the obvious symptoms could only be observed for 3 to 4 days (Figure 4D) before the piglets quickly recovered (5 to 7 days), but viral nucleic acid could be detected in the feces of infected pigs for a long period of time (Figure 7A). This might be the reason why most reports on the isolation of EV-Gs do not depict the skin symptoms in the pigs from which the sample is derived. As is the case in this study, no obvious skin symptoms were observed in the pigs from which the EV-G-positive fecal samples were collected. 

Massive epidemiological investigations have revealed that there is no statistically significant difference in the prevalence rates of EV-G infection in the pigs with or without diarrhea [2,3,7]. Similarly, PLCP-inserted EV-Gs had been detected in the feces of pigs with diarrhea in the USA, Belgium, China, Germany, South Korea, and Japan [16,18,19,24,43,44], while PLCP recombinant EV-Gs had been identified from normal feces of pigs in Japan [2,18,42], and the association between EV-G-PLCP strain infection in pigs and diarrhea is still unknown. According to the results of this study, EV-G/YN23/2022 (EV-G17-PLCP) infection did cause diarrhea in piglets (Figure 4C). Throughout the infection experiment (21 days), a high load of viral nucleic acid in the feces (Figure 7A) and small intestine (Figure 7B) were detected, indicating that the intestinal tract was an important target organ for viral infection, and the virus had proliferated in the digestive tract and was shed in feces for a long period of time. This might partially explain the high EV-G nucleic acid positive rates in many EV-G epidemiology surveys [7,20,45]. The results of the histopathological examination showed that EV-G/YN23/2022 infection could cause severe lesions in the intestinal tissues, including small intestinal epithelial cell necrosis and intestinal villi atrophy (Figure 6A). Taken together, the small intestine was an important target organ for EV-G/YN23/2022 infection, and virus infection caused severe tissue damage in this target organ; hence, we propose that these effects may be the main triggers of diarrhea in piglets caused by EV-G/YN23/2022 infection. 

EV-G is a member of the family *Picornaviridae*, and some of the viruses of this family can infect the central nervous system (CNS) of pigs, such as porcine encephalomyocarditis virus (EMCV) [46], porcine teschovirus (PTV) [47], and porcine sapelovirus (PSV) [48]. EV-G has also been reported to invade the CNS of pigs and cause neurological symptoms, manifesting as hind limb trembling or paralysis and aberrant excitatory [21,22]. In this study, a high virus load (Figure 7B) and strong viral RNA-positive signals (Figure 6B) were detected in the brains of the EV-G/YN23/2022-inoculated pigs, indicating that EV-G/YN23/2022 had invaded the brains of the piglets. However, no neurological symptoms were observed in the infected piglets during the entire experimental period, and pathological lesions were also not observed in the pig brain tissues, which is inconsistent with the results of previous reports [21,22]. We speculate that the difference in pathogenicity of the different EV-G strains or the difference in the pig breeds used for the infection trials may contribute to these discrepancies regarding clinical symptoms. In fact, it was not surprising that different virus strains displayed different pathogenicity profiles; the same situation has also been observed for many other pathogens in pigs, such as porcine reproductive and respiratory syndrome virus (PRRSV) [49], porcine circovirus (PCV) [50], pseudorabies virus (PRV) [51], and porcine epidemic diarrhea virus (PEDV) [52]. Nevertheless, in addition to the different viral strains and pig breeds, the severity of symptoms caused by infection is likely to be related to the age of the piglets, for example, the concentrations of cytokines in the serum of weaned piglets were higher than those of suckling piglets [53], which could explain why we did not observe lesions in brains of EV-G17-PLCP-inoculated 31-day-old piglets (Figure 5 and Figure 6A) compared with the reports of Yang et al. (14-day-old piglets were used) [21] and Xue et al. (7-day-old piglets were used) [22]. 

EV-G/YN23/2022 propagated effectively and caused strong CPEs in BHK-21, Marc-145, PK-15, ST, and SIEC cells (Figure 1A), which is in agreement with the reports that EV-Gs could propagate in BHK-21, Vero, ST, and Marc-145 cells [1,22,24,54]. The growth curves of EV-G/YN23/2022 showed that the virus propagated the most rapidly in SIEC cells and reached a peak (7.2 × 10^6^ PFU/mL) at 48 hpi (Figure 1C), which was higher than the virus titers for the other cell lines (Figure 1C). Shang et al. demonstrated that the PLCP knockout EV-G showed impaired growth in ST cells when compared to the parent virus (08/NC_USA/2015, EV-G17) [24], suggesting that the PLCP gene may affect virus proliferation. Xue et al. reported that CH/17GXQZ/2017 (EV-G1, without PLCP) showed a relatively long replication cycle in Marc-145 cells and reached its peak (5 × 10^6^ PFU/mL) at 72 hpi [22], which was markedly slower than that of EV-G/YN23/2022 (with PLCP) (Figure 1C). Although the differences among the different EV-G strains or genotypes cannot be excluded, it may mirror the stronger proliferation capacity of EV-G-PLCP strains when compared to that of EV-Gs without the PLCP gene. Generally, EV-G/YN23/2022 showed strong tropism to SIEC cells, with excellent replication ability, which may be a trigger that leads to severe cytopathic changes in the small intestine and, ultimately, diarrhea after EV-G/YN23/2022 infection. 

Shang et al. demonstrated that the EV-G-PLCP strain was able to inhibit the expression of innate immune genes in ST cells [24], but the immune responses of the natural host are still unknown. According to our study, the piglets showed a strong inflammatory response after oral inoculation of EV-G/YN23/2022, with a significant elevation in the mRNA levels of IL-6, IL-18, IFN-α, and IFN-β (Figure 8). The variation trends of IL-6 concentration in serum and its transcription level were basically consistent from 6 to 12 dpi (Figure 8 and Figure 9), and the excessive expression of IL-6 could intensify the intestinal permeability and further lead to diarrhea [55], which was in line with the phenomenon of piglets’ diarrhea from 6 to 9 dpi and the results of histopathological analysis. These results indicated that the dramatic increase in expression of cytokine IL-6 may be closely related to piglet diarrhea. Admittedly, there is no denying that IL-1β, TNF-α and IL-18 exhibit nonnegligible contributions to fever, vascular permeability, and ion transport in the intestinal epithelial cells of piglets [56,57], and all these pro-inflammation cytokines together caused mild hemorrhaging in the small intestine, intestinal villi atrophy, local exfoliation, and defects of the epithelial mucosae (Figure 5 and Figure 6). However, there are differences in the expression levels of cytokines between serum and intestinal tissue [57], and moreover we did not detect significant increases of the other pro-inflammation cytokines in the serum of challenged piglets (Figure 9), therefore we could only draw an inference that the dramatic increase in expression of cytokine IL-6 in piglet serum may be closely related to diarrhea and a potential biomarker associated with EV-G infection.

IFN-λ3 plays a critical role against infection, particularly in mucosal infection of the gastrointestinal tract [58]. The previous study also reported that the translation level of IFN-β was up-regulated in ST cells post-EV-G infection [24]. In addition, the concentrations of IFN-α and IFN-β in serum were elevated to a different extent from 12 to 15 dpi and from 9 to 15 dpi, respectively, along with the increase of IFN-λ3 from 6 to 15 dpi, although this was not significant (Figure 9). Meanwhile, the surge in expression level of the antiviral gene ISG-15 was evidence for the function of IFNs (Figure 8) [59]. Hence, we posit that as the innate immune system’s first line of defense, the IFN system may play an important role in the antiviral activity of in vivo and in vitro EV-G infections. The elevation of the mRNA level of IL-28B (IFN-λs) was detected in ST cells [24]; however, it stayed at a constant level in the PBMCs in this study (Figure 8). Only single-type cells (ST cells) were used in the in vitro research, while the hosts’ immune systems were extremely complex in the in vivo tests and were regulated by a variety of elements. Therefore, it is possible that there are inconsistencies regarding the IFN-λ3 transcription levels between in vivo and in vitro assays. 

Previous studies pointed out that inoculation of EV-G strains induced clinical and pathological symptoms in suckling piglets [22,29], however, there was still a lack of research on the pathogenicity of PLCP-insertional and non-PLCP-insertional strains belonging to the same genotype in animals, and additionally we did not isolate EV-G17 strains without PLCP gene insertion in our study, which made it difficult to adequately compare and analyze the effects of PLCP insertion on disease progression, immune responses, and overall disease outcomes. In order to determine the induction of pro-inflammation cytokines and the intestinal damage caused by PLCP gene inserted EV-Gs on weaned piglets more clearly, the EV-G17-PLCP strain is being modified by reverse genetic technology into a non-PLCP-insertional strain, and corresponding animal pathogenicity experiments as well as the detection of intestinal pro-inflammation cytokines of piglets, are planned to be conducted. Furthermore, PBMC-stimulated assays and flow cytometry of immune cells, should also be carried out to reveal differences of host immune responses against EV-G17-PLCP and non-PLCP-insertional strains infections in vitro.

## Figures and Tables

**Figure 1 viruses-15-01747-f001:**
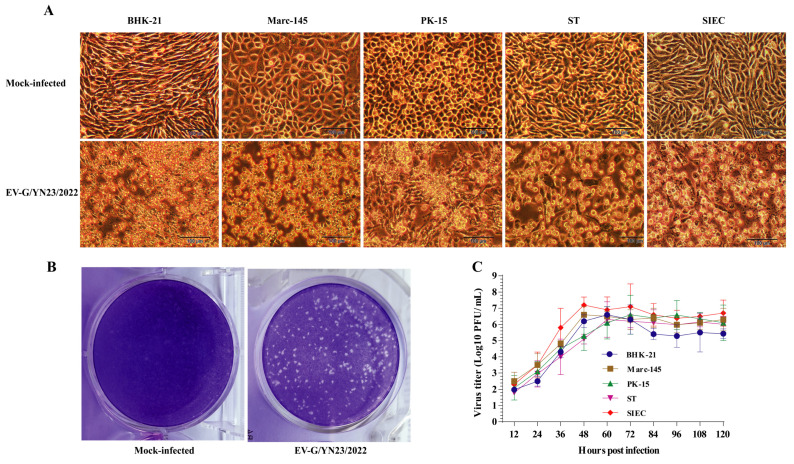
The CPEs, plaque formation, and growth curves of the EV-G/YN23/2022 strain. (**A**) The CPEs of EV-G/YN23/2022 in various cell lines (scale bar = 100 μm). (**B**) Plaque formation of EV-G/YN23/2022 in BHK-21 cells (stained with crystal violet at 72 hpi). (**C**) Growth kinetics of EV-G/YN23/2022 in different cell lines with an MOI of 0.01.

**Figure 2 viruses-15-01747-f002:**
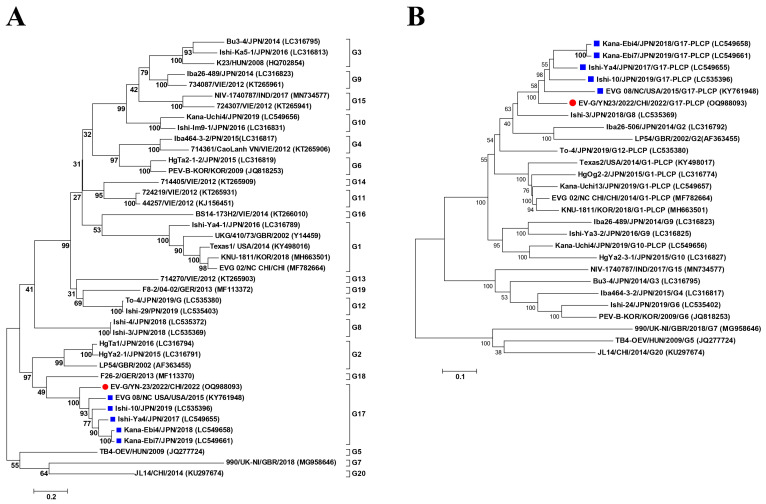
Phylogenetic analysis based on the nucleotide sequences of the VP1 gene (**A**) and full-length genomes (**B**) of EV-G strains. The tree was constructed by the maximum likelihood method using MEGA 6.06 with the best-fitting model ‘GTR+G+I’. The numbers on the branches are bootstrap values (percent) from 1000 replicates. The EV-G/YN23/2022 strain isolated in this study is marked with a red dot, and the previously published EV-G17 strains are labeled with a blue box. The sequences of other representative EV-Gs are indicated as ‘Strain/Country/Year/(GenBank accession number)’.

**Figure 3 viruses-15-01747-f003:**
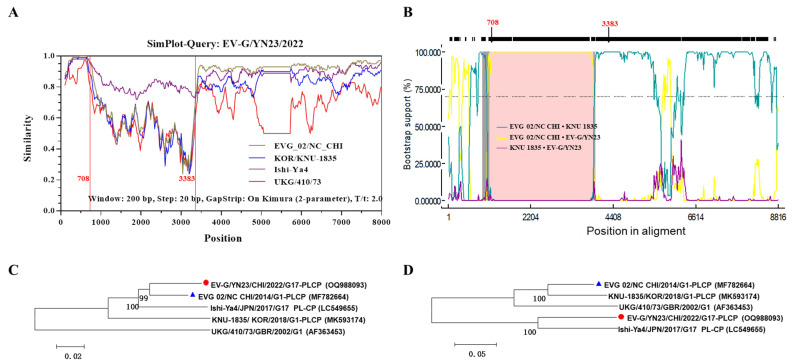
Genome recombination analyses of EV-G/YN23/2022. (**A**) Similarity plot analysis of the genome sequences of EVG 02/NC_CHI (G1) (brown curve), KNU-1835 (G1) (blue curve), Ishi-Ya4 (G17) (purple curve), UKG/410/73 (G1) (red curve), and EV-G/YN23/2022 as query sequences; the y-axis presents the percent identity with a window size of 200 bp and a step size of 20 bp. The vertical line indicates the regions where a recombination event may occur. (**B**) The recombination breakpoint analysis of EVG 02/NC CHI (G1) vs. KNU-1835 (G1) (light green curve), EVG 02/NC CHI (G1) vs. EV-G/YN23 (G17) (yellow curve), and KNU 1835 (G1) vs. EV-G/YN23 (G17) (purple curve). (**C**) The phylogenetic tree constructed by the neighbor-joining method in MEGA 6.06 based on the concatenated regions (positions 1–708 and 3383–8033) of the EV-G/YN23/2022 genome. (**D**) The phylogenetic tree constructed by the neighbor-joining method in MEGA 6.06 based on the 709–3382 region of the EV-G/YN23/2022 genome.

**Figure 4 viruses-15-01747-f004:**
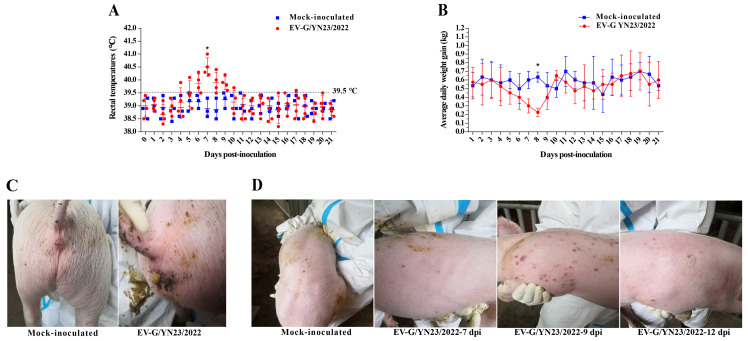
The rectal temperature, weight gain, and clinical symptoms of EV-G/YN23/2022-infected piglets and mock-inoculated controls. (**A**) Rectal temperatures of piglets infected with EV-G/YN23/2022 and mock-inoculated piglets; mean ± SD temperature (°C) is shown, and the fever cut-off value was set at 39.5 °C. Statistical analyses were performed using the *t*-test. * *p* < 0.05. (**B**) Average daily weight gain of the piglets inoculated with EV-G/YN23/2022 and mock-inoculated piglets; mean ± SD weight gain (kg) are shown. Statistical analyses were performed using the *t*-test. * *p* < 0.05. (**C**) Mild diarrhea observed from EV-G/YN23/2022-challenged piglets at 8 dpi. (**D**) Skin rashes of the EV-G/YN23/2022-challenged piglets at 7, 9, and 12 dpi.

**Figure 5 viruses-15-01747-f005:**
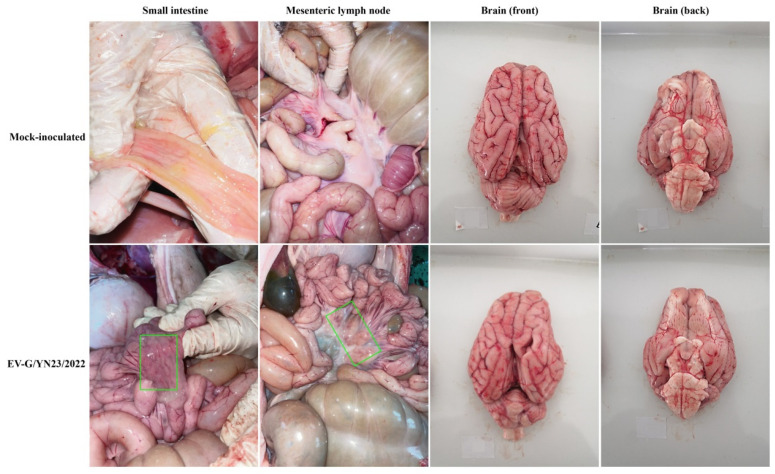
Gross lesions of piglets challenged with EV-G/YN23/2022 and mock-inoculated piglets. Mild hemorrhaging in the small intestine and mesenteric lymph node (green rectangle) of the challenged animals were observed.

**Figure 6 viruses-15-01747-f006:**
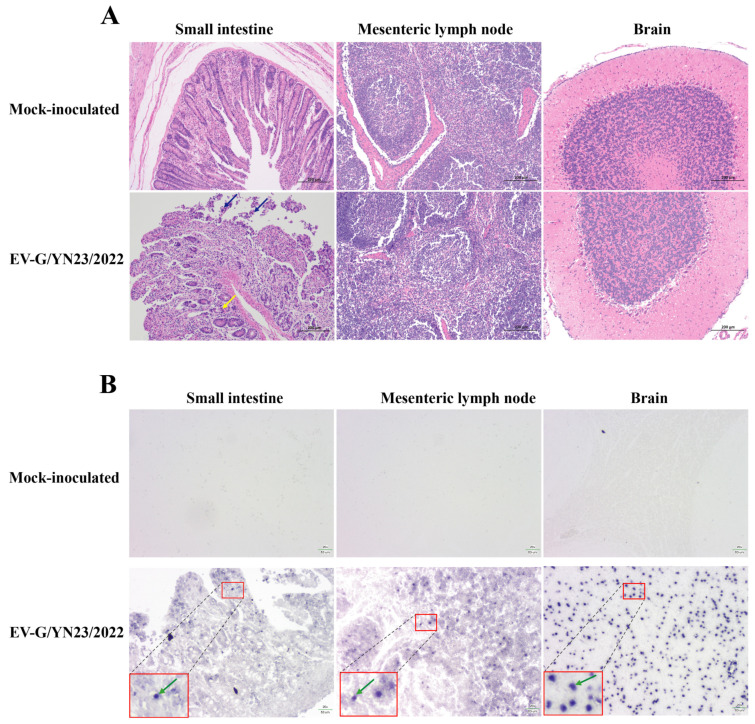
Histopathological analysis of the piglets challenged with EV-G/YN23/2022 and mock-inoculated controls. (**A**) Hematoxylin and eosin (H&E) staining of the small intestine showed local exfoliation and defects in the epithelial mucosae (blue arrow), and the irregularly arranged and enlarged space of the lamina propria (yellow arrow) in the infected pigs when compared to the control group (scale bar = 200 μm). (**B**) In situ hybridization showing the distribution of viral nucleic acids in the small intestine, mesenteric lymph node, and brain (cerebellum) tissue sections (green arrow) of the piglets challenged with EV-G/YN23/2022, and mock-inoculated piglets showed no signal (scale bar = 50 μm).

**Figure 7 viruses-15-01747-f007:**
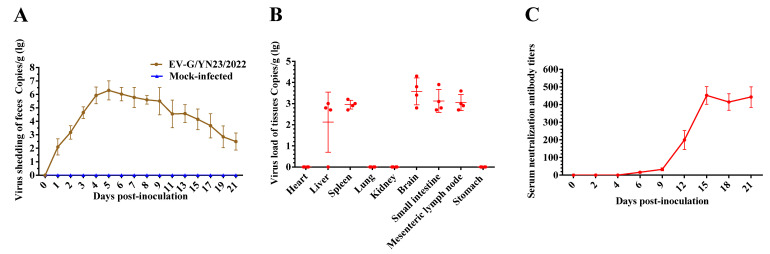
Viral RNA copies in samples (fecal swab, heart, liver, spleen, lung, kidney, brain, small intestine, mesenteric lymph node, and stomach) and the serum neutralization antibody titers of the piglets challenged with EV-G/YN23/2022. (**A**) Fecal virus shedding in EV-G/YN23/2022-challenged piglets from 0 to 21 dpi. (**B**) Virus loads in different organs of EV-G/YN23/2022-challenged piglets at 21 dpi. (**C**) nAbs against EV-G/YN23/2022 of challenged piglets; the serum neutralization titer was defined as the highest serum dilution inhibiting 50% of virus growth.

**Figure 8 viruses-15-01747-f008:**
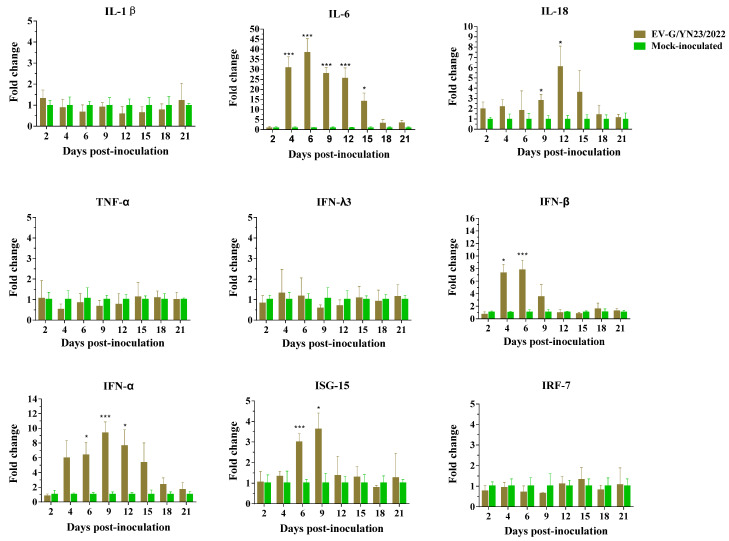
The expression profiles of multiple immune-related cytokine genes (IL-1β, IL-6, IL-18, TNF-α, IFN-α, IFN-β, IFN-λ3, ISG-15, and IRF-7) in the PBMCs of the EV-G/YN23/2022-inoculated piglets at 2, 4, 6, 9, 12, 15, 18, and 21 dpi. The relative expression levels of the target genes were normalized to the β-actin gene and calculated using the 2^−ΔΔCt^ method. The y-axis represents the fold change of the target gene expression in the experimental group vs. that of the control group. Statistical analyses were performed using the *t*-test. * *p* < 0.05; *** *p* < 0.001.

**Figure 9 viruses-15-01747-f009:**
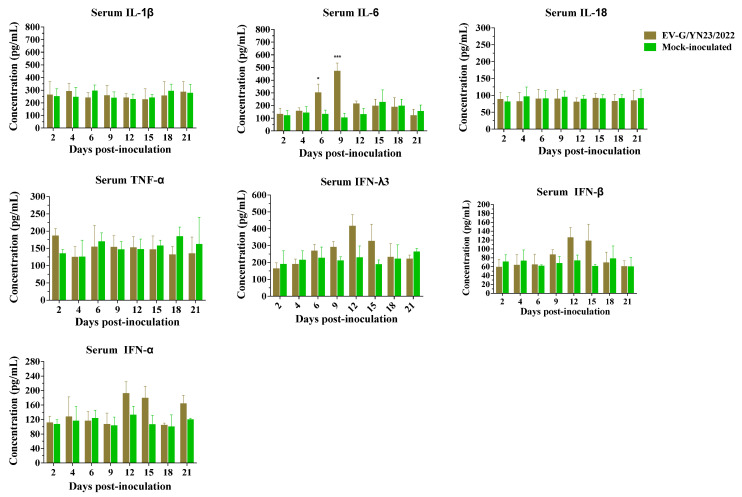
The protein concentration of IL-1β, IL-6, IL-18, TNF-α, IFN-α, IFN-β, and IFN-λ3 in the serum of the piglets challenged by EV-G/YN23/2022. Statistical analyses were performed using the *t*-test. * *p* < 0.05; *** *p* < 0.001.

**Table 1 viruses-15-01747-t001:** Primers used for amplifying the complete genome sequences of EV-G isolate and RT-qPCR amplification.

Primer	Sequence (5′⟶3′)	Position ^a^	Product Length (bp)
EV-G-F1	TTAAAACAGCCTGTGGGTTG	1–20	2058
EV-G-R1	TCCACACAGGGTTTTGGACAT	2038–2058	
EV-G-F2	AGATGAGGATAGCAGCGCATGT	1806–1827	3016
EV-G-R2	CTCAGCCACTTCAATATCCACA	4800–4821	
EV-G-F3	TGGTGATGGATGACCTTAATCAG	4583–4605	1908
EV-G-R3	AGAACGACTCCACCACACTGTCC	6469–6491	
EV-G-F4	CGACCGTTCTATCCAGCTTAGT	5888–5909	2159
EV-G-R4	GTACACCCCATCCGGTGGGTGTAT	8023–8046	
EV-G-SG-F	CTGAATGCGGCTAATCCTAAC	492–512	107
EV-G-SG-R	AACACGGACACCCAAAGTA	580–598	

^a^ The positions of primers EV-G-F1/R1 to F4/R4 were determined based on the representative strain EVG 02/NC_CHI/2014 (MF782664); the positions of primers EV-G-SG-F/R were determined based on EV-G/YN23/2022 (OQ988093).

**Table 2 viruses-15-01747-t002:** Nucleotide and amino acid identity (%) of EV-G/YN23/2022 with reference EV-G strains.

EV-G/YN23/2022 Gene or Protein Region (nt/aa)	G17-PLCP (LC535396)	G17-PLCP (LC549655)	G17-PLCP (LC549658)	G17-PLCP (LC549661)	G17-PLCP (KY761948)	G1-PLCP (MF782664)	G1-PLCP (KY498017)	G8 (LC535369)	G7 (MG958646)
5′UTR (812)	NA ^a^	92.5	91.8	92	94.9	**95.6** ^b^	84.5	84.7	50.4
3′UTR (66)	89.1	90.4	NA ^a^	**92.4**	89.1	87.8	NA ^a^	87.8	78.6
Complete genome/ polyprotein (8033/2385)	NA ^a^/95.8	**87.5**/**96.6**	NA ^a^/95.3	85.7/95.4	85.5/95.3	84.5/88.5	NA ^a^/87.4	76.8/82.5	62.5/73.1
1A/VP4 (207/69)	82.6/**100.0**	**89.8**/**100.0**	87.9/98.5	87.9/98.5	87.4/97.1	79.7/91.3	75.3/85.5	79.2/86.9	74.8/85.5
1B/VP2 (738/246)	80.8/96.7	81.8/98.3	81.0/97.5	82.1/97.5	**82.9**/**98.7**	69.1/83.3	70.8/84.5	74.9/88.2	66.1/76.0
1C/VP3 (840/280)	81.7/93.5	82.2/95.3	80.5/95.3	81.1/95.7	**84.1**/**95.7**	67.9/71.7	66.6/71.4	71.1/76.7	59.2/64.6
1D/VP1 (729/243)	82.5/97.5	82.8/97.9	82.8/97.9	**83.1**/**98.3**	82.5/96.7	60.5/62.5	60.6/62.5	66.1/70.3	60.5/58.6
2A/Pro (450/150)	84.0/96.0	84.4/94.6	84.8/95.3	84.8/95.3	84.6/95.3	**92.6**/**96.6**	85.1/94.6	92.2/95.3	73.5/78.6
2B (297/98)	83.8/96.9	87.5/94.8	82.1/96.9	82.4/96.9	88.2/95.9	**91.5**/**97.9**	83.1/94.8	89.8/96.9	69.3/81.6
2C/ATPase (987/329)	87.8/97.5	88.7/98.4	83.7/95.4	84.4/95.7	86.0/96.9	**91.9**/97.8	85.7/96.6	90.3/**98.1**	75.8/88.7
PLCP (642/214)	82.9/83.4	85.0/**86.9**	76.8/76.6	77.0/77.1	76.5/79.0	**85.8**/85.2	80.2/79.8	NA ^c^	NA ^c^
3A (267/89)	89.5/97.7	88.0/96.6	85.0/**100.0**	86.8/**100.0**	85.0/98.8	**94.0**/**100.0**	86.8/96.6	91.0/**100.0**	74.1/85.3
3B/VPg (66/22)	83.3/**100.0**	86.3/**100.0**	92.4/**100.0**	83.3/**100.0**	93.9/**100.0**	89.3/**100.0**	89.3/**100.0**	95.4/**100.0**	75.7/90.9
3C/Pro (549/183)	88.8/97.8	90.8/**98.9**	89.7/**98.9**	89.4/**98.9**	85.6/97.2	**92.5**/98.3	89.0/**98.9**	**92.1**/**98.9**	77.7/90.7
3D/Pol (1383/461)	91.9/98.4	91.7/98.2	92.2/98.0	91.7/97.8	86.8/96.7	**94.2**/**98.4**	90.6/98.2	90.0/97.6	79.5/91.1

^a^ NA: not applicable. ^b^ Bold face numbers depict the highest identity. ^c^ No PLCP gene in LC535369 and MG958646.

## Data Availability

The complete genome sequence of EV-G/YN23/2022 obtained in this study have been deposited in the GenBank under the accession number OQ988093.

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
