# Peer review of "First Identification and Pathogenicity Evaluation of an EV-G17 Strain Carrying a Torovirus Papain-like Cysteine Protease (PLCP) Gene in China"

_viruses, 2023, doi:10.3390/v15081747_

Round 1

Reviewer 1 Report

This study is generally well-conducted and presents valuable insights into the topic. With attention to and resolution of the comments provided, I believe the manuscript would be suitable for acceptance.

This study presents the inaugural detection of EV-G17 in China. An in-depth phylogenetic analysis and recombination study were conducted to delve into the intricacies of the viral genome. Moreover, animal inoculation experiments revealed that EV-G17 induces distinct skin lesions alongside the expected diarrhea symptoms. The findings would have been further enriched had there been a comparison with a control group inoculated with an EV-G strain devoid of the PLCP gene.

  • Line 160-163: Kindly include a supplementary CSV file listing the strain names and their respective accession numbers. This will significantly aid subsequent research endeavors.
  • Line 167-170: Could you elucidate the reasoning behind adopting the "GTR+G+I" model? Were there any comparative analyses with alternative models during the selection process? Please specify the methods or criteria employed for model selection.
  • Line 225-227: Please provide details regarding the beads utilized for sample pulverization.
  • Line 298-300: You noted the observation of clear and uniform plaques during the fourth day. Could you describe the plaque morphology on days 1-3?
  • Table 2: The table contains unnatural line breaks, compromising its readability. Consider reducing the font size of the accession numbers or adjusting the layout to ensure each entry remains on a single line. The narrow column gaps make differentiation challenging; please expand the spacing between columns.
  • Line 326-328: Could you elaborate on the criteria for selecting the nine strains as representative sequences of EV-G?
  • Line 328-332: Please modify "1D gene" to "1D/VP1 gene" for clarity. This adjustment will enhance the coherence between the text and the accompanying table.
  • Figure 2: The text within the phylogenetic tree could be more representative, making it more discernable. A layout modification to enhance legibility would be appreciated.
  • Figure 3: The font size is notably tiny, affecting readability.
  • Figure 4: The body temperature graph's font size is insufficient. A size increment is recommended.
  • Line 423-424: Please specify the brain region utilized for ISH, which result is depicted in Figure 6.
  • Figure 6: The epithelial mucosa loss, indicated by the blue arrow, is not discernible in the provided image. While the irregular layering highlighted by the yellow arrows is evident, the spacing doesn't seem notably expanded.
  • Figure 7: The font size should be increased for better readability.
  • Figure 8: Given that the control group consistently registers at 1, you might consider omitting it from the graph for clarity. If retained, please incorporate error bars to represent variation. Additionally, an increase in font size is suggested for improved legibility.
  • Line 632-638: It may be reasonable to link diarrhea to high levels of IL-6, but it's important to consider other cytokines' potential involvement and interactions. It's possible that a combination of cytokines could have contributed to intestinal inflammation and higher permeability, resulting in diarrhea. The authors should acknowledge the potential role of other cytokines in their analysis.

Author Response

Dear Editors and Reviewers:

Thank you very much for your positive and constructive comments and suggestions on our manuscript entitled “First Identification and Pathogenicity Evaluation of an EV-G17 Strain Carrying a Torovirus Papain-like Cysteine Protease (PLCP) Gene in China”. (ID: viruses-2547187). Those comments are all valuable and very helpful for revising and improving our paper, as well as the important guiding significance to our researches. We have studied comments carefully and have made correction which we hope meet with approval. Revised portion are marked in red in the paper. The main corrections in the paper and the responds to the reviewer’s comments are as flowing:

Point 1: Line 160-163: Kindly include a supplementary CSV file listing the strain names and their respective accession numbers. This will significantly aid subsequent research endeavors.

Response 1: Line 164: ‘Table S1. Information on the 47 reference EVGs downloaded from GenBank.’ was added into the supplementary materials.

Point 2: Line 167-170: Could you elucidate the reasoning behind adopting the "GTR+G+I" model? Were there any comparative analyses with alternative models during the selection process? Please specify the methods or criteria employed for model selection.

Response 2: Line 171-172: The appropriate best-fit model was selected using MEGA6.06 based on the Bayesian Information Criterion (BIC), and the model with the lowest BIC scores was considered the best-fit model.

Point 3: Line 225-227: Please provide details regarding the beads utilized for sample pulverization.

Response 3: Line 231-232: The beads utilized for homogenizing samples were stainless steel beads with 2 mm in diameter (EASYBIO, Beijing, China).

Point 4: Line 298-300: You noted the observation of clear and uniform plaques during the fourth day. Could you describe the plaque morphology on days 1-3?

Response 4: (1) The clear and uniform plaques presented in Figure 1B were observed on the BHK-21 cells at 4 days post infection (dpi), and the virus was purified through four rounds of plaque purification. (2) Non CPE was observed at 1 dpi, and a few cells turned round on day 2, and there were more cells turned round and a few cells were lysed on day 3, and massive cells were lysed and clear plaques were observed at 4 dpi.

Point 5: Table 2: The table contains unnatural line breaks, compromising its readability. Consider reducing the font size of the accession numbers or adjusting the layout to ensure each entry remains on a single line. The narrow column gaps make differentiation challenging; please expand the spacing between columns.

Response 5: We have reduced the fond size of the texts in the table 2, and the gaps between columns were expanded slightly.

Point 6: Line 326-328: Could you elaborate on the criteria for selecting the nine strains as representative sequences of EV-G?

Response 6:

(1) The sequences LC535396, LC549655, LC549658, LC549661 and KY761948 were the all EV-G17 complete genomes that had been deposited on the GenBank database; (2) The sequences MF782664 and KY498017 were the EVG-PLCP strains sharing the highest and lowest genome sequence identity to EV-G/YN23/2022, respectively; (3) The sequences LC535369 and MG958646 are the EVG strains (non-PLCP) sharing the highest and lowest genome sequence identity to EV-G/YN23/2022, respectively.

Point 7: Line 328-332: Please modify "1D gene" to "1D/VP1 gene" for clarity. This adjustment will enhance the coherence between the text and the accompanying table.

Response 7: Line 336: We have modified "1D gene" to "1D/VP1 gene" in the manuscript.

Point 8: Figure 2: The text within the phylogenetic tree could be more representative, making it more discernable. A layout modification to enhance legibility would be appreciated.

Response 8: (1) We have removed the redundant information, simplified the sequence names as ‘Strain/Country/Year/(GenBank accession number)’. (2) There are much more available VP1 sequences on the GenBank database than the whole genome sequences of EVGs, and in order to display the phylogenetic relationships of different genotypes of EVGs comprehensively, we selected more VP1 sequences to construct the phylogenetic tree than the whole genome sequences, so the layout was not modified.

Point 9: Figure 3: The font size is notably tiny, affecting readability.

Response 9: We have magnified the font size.

Point 10: Figure 4: The body temperature graph's font size is insufficient. A size increment is recommended.

Response 10: The font size of the body temperature graph and average daily weight gain graph has been increased.

Point 11: Line 423-424: Please specify the brain region utilized for ISH, which result is depicted in Figure 6.

Response 11: The brain region utilized for ISH is cerebellum.

Point 12: Figure 6: The epithelial mucosa loss, indicated by the blue arrow, is not discernible in the provided image. While the irregular layering highlighted by the yellow arrows is evident, the spacing doesn't seem notably expanded.

Response 12: We replaced the original image with the better image, and re-marked the location of epithelial mucosal loss (blue arrow), the description of expanded space of lamina propria was deleted.

Point 13: Figure 7: The font size should be increased for better readability.

Response 13: We have magnified the font size of Figure 7.

Point 14: Figure 8: Given that the control group consistently registers at 1, you might consider omitting it from the graph for clarity. If retained, please incorporate error bars to represent variation. Additionally, an increase in font size is suggested for improved legibility.

Response 14: The error bars of the control group have been added and the font size has been magnified.

Point 15: Line 632-638: It may be reasonable to link diarrhea to high levels of IL-6, but it's important to consider other cytokines' potential involvement and interactions. It's possible that a combination of cytokines could have contributed to intestinal inflammation and higher permeability, resulting in diarrhea. The authors should acknowledge the potential role of other cytokines in their analysis.

Response 15: The results of our study indicated that the dramatic increase in expression of cytokine IL-6 may be closely related to piglet diarrhea. Admittedly, there is no denying that IL-1β, TNF-α and IL-18 exhibit nonnegligible contributions to fever, vascular permeability and ion transport in the intestinal epithelial cells of piglets [56,57], and all these pro-inflammation cytokines together caused mild hemorrhaging in the small intestine, intestinal villi atrophy and local exfoliation and defects of the epithelial mucosae (Figure 5,6). However, there are differences in the expression levels of cytokines between serum and intestinal tissue [57], and moreover we did not detect significant increases of the other pro-inflammation cytokines in serum of challenged piglets (Figure 9), therefore we could only draw an inference that the dramatic increase in expression of cytokine IL-6 in piglet serum may be closely related to diarrhea and a potential biomarker associated with EV-G infection. In the further studies, we would investigate the cytokine profile of different organs to gain insights into the unique effects of EV-G17-PLCP on these organs.

Reviewer 2 Report

The findings of this study are particularly intriguing for readers in the veterinary medicine field. However, to further enhance the understanding of the role of PLCP in the pathogenesis of EV-G17, the authors received valuable feedback from reviewers that highlights three major comments for improvement.

The authors are encouraged to compare the characteristics of EV-G17-PLCP with those of EV-G17 (no PLCP) strains, both in vitro and in vivo assays. This comparative analysis could shed light on the specific contributions of PLCP to the virus's behavior and pathogenicity. 

In-depth comparison of pathogenic changes in both the brain and small intestine is suggested. This includes a comprehensive investigation of cytokine profiling to gain insights into the unique effects of EV-G17-PLCP on these organs. 

The authors should explicitly discuss the significance of PLCP in the pathogenesis of EV-G17. Emphasizing the role of PLCP in disease progression, immune responses, and overall disease outcomes would provide valuable insights into potential therapeutic targets and control strategies.

 Addressing these major comments will undoubtedly enrich the study's impact and further contribute to the veterinary medicine community's understanding of porcine torovirus infections.

The English quality in this paper is satisfactory.

Author Response

Dear Editors and Reviewers:

Thank you very much for your constructive comments and suggestions on our manuscript entitled “First Identification and Pathogenicity Evaluation of an EV-G17 Strain Carrying a Torovirus Papain-like Cysteine Protease (PLCP) Gene in China”. (ID: viruses-2547187). Those comments are very valuable and helpful for revising and improving our paper, as well as the important guiding significance to our researches. We have studied comments carefully and have made correction which we hope meet with approval. Revised portion are marked in red in the paper. The main corrections in the paper and the responds to the reviewer’s comments are as flowing:

Point 1: The authors are encouraged to compare the characteristics of EV-G17-PLCP with those of EV-G17 (no PLCP) strains, both in vitro and in vivo assays. This comparative analysis could shed light on the specific contributions of PLCP to the virus's behavior and pathogenicity.

Response 1:

(1) The characteristics of EV-G17-PLCP and EV-G17 (no PLCP) strains were compared comprehensively in vitro assays by Shang et al. [42].

(2) Admittedly, EV-G17 (no PLCP) strain was not isolated in our study, which made it impossible for us to conduct the comparative assay in vivo.

(3) To our knowledge, currently all EV-G17 isolates in the world were the PLCP-insertion strains, and our report was the first research about the pathogenicity experiment of EV-G17 PLCP strain, therefor we can’t find any available comparative data about the pathogenicity of EV-G17 PLCP strain. (4) As an alternate approach, we compared our results with the other previous reports, even though the EV-G strains they used were neither non-PLCP strain nor EV-G17 strain.

Point 2: In-depth comparison of pathogenic changes in both the brain and small intestine is suggested. This includes a comprehensive investigation of cytokine profiling to gain insights into the unique effects of EV-G17-PLCP on these organs.

Response 2:

(1) Neither any neurological symptoms nor pathological lesions were observed in the infected piglets during the entire experimental period, so we didn’t detect the cytokine profiles of the piglet brain in our study.

(2) We have comprehensively analyzed the possible association between various cytokines in the peripheral blood and the small intestine lesions, and found that small intestine lesions and diarrhea of the EV-G17-PLCP infected piglets may be the combination results of multiple cytokines, while, IL-6 may play an important role.

(3) In order to investigate the complete disease progression, we dissected the animals until 21 days post inoculation. Considering that the expression levels of all cytokines in peripheral blood have returned to the constant levels, we speculated that the cytokine expression levels in the small intestine may have also returned to constant, therefore, we did not detect cytokines in the small intestine at 21 dpi.

Point 3: The authors should explicitly discuss the significance of PLCP in the pathogenesis of EV-G17. Emphasizing the role of PLCP in disease progression, immune responses, and overall disease outcomes would provide valuable insights into potential therapeutic targets and control strategies.

Response 3: Constructing the gene knockout recombinant virus by reverse genetics, which was an important method to explore the function of the specific gene. In order to determine the significance of PLCP gene in the pathogenesis of EV-G17, the PLCP-knockout EV-G17 strain would be constructed, and a series of compared assays would be conducted in vitro and in vivo to reveal the role of PLCP gene in disease progression, immune responses, and overall disease outcomes, especially the intestinal cytokine expression profiles induced by the wild-type EV-G17-PLCP strain and the PLCP knockout mutant would be compared meticulously to clarify the correlation between cytokines and intestinal lesions and diarrhea.

Round 2

Reviewer 2 Report

None.